

# The water temperature characteristics of the Lena River at basin outlet in the summer period

Vera Fofonova[1], Igor Zhilyaev[2], Marina Kraineva[3], Dina Iakshina[3], Nikita Tananaev[4], Nina Volkova[5], Karen H. Wiltshire[1]

[1]Alfred Wegener Institute, Helmholtz Centre for Polar and Marine Research, List, 25992, Germany
[2]Southern Scientific Centre of Russian Academy of Sciences, Rostov-on-Don, 344006, Russia
[3]Institute of Computational Mathematics and Mathematical Geophysics, Siberian Branch Russian Academy of Sciences, Novosibirsk, 630090, Russia
[4]Igarka Geocryology Laboratory, Permafrost Institute, Siberian Branch Russian Academy of Sciences, Igarka, 663200, Russia
[5]State Hydrological Institute, 23, St. Petersburg, 199053, Russia

*Correspondence to*: Vera Fofonova (vera.fofonova@awi.de)

**Abstract.** The water temperature characteristics of the Lena River at basin outlet during the summer season (June–September) are considered. The analysis is based on a long-term data series covering the period from the beginning of observation (1936) to the present time (2012) at Kusur (Kyusyur) gauging station and complementary data at several stations downstream and one station upstream. These additional data are rarely used, but their analysis is important for understanding processes in the basin outlet area. The differences between the stream surface temperatures at Kusur station and 200 km downstream to the north at Habarova (Khabarova) station have almost always been an anomalously large and negative for the considered period since the beginning of observation during open water season from July to September. The description of this difference and its analysis are presented. To sort the problem out, we consider the observational data in terms of the hydrology and morphology of the Lena River delta and main channel area and apply statistical and deterministic modelling approaches. The inability of water temperature observational data at Kusur station to represent the mean cross-sectional temperature is addressed. The analysis of the water temperature trends at the Kusur and Habarova stations is also presented.

## 1 Introduction

The Lena River is one of the largest rivers in the Arctic with the largest delta. Permafrost underlies 78–93 % of the watershed, with continuous permafrost extending south to 50° N (Zhang et al., 1999). Observational data available for the Lena River suggest an on-going change in climate and biological factors over the last 50 years (e.g. Kraberg et al., 2013; McClelland et al., 2006; Yang et al., 2002). Costard et al. 2007 found that the Lena water temperature in the flood period had increased at Tabaga station by up to 2 °C, as compared to the values in 1950, and that this increase had contributed to coastal erosion and modified the chemical water composition. Most biological communities and species are very sensitive to changes in water temperature and water chemistry (Conlan et al. 2005; Kraberg et al., 2013). Restructuring of an ecosystem





may follow such changes. Water mass characteristics at the Lena River basin outlet are particularly important for dynamics of the Laptev Sea and the Arctic Ocean as a whole (e.g. Dmitrenko et al., 2008; Morison et al., 2012; Yang et al., 2005). The Lena River Delta has a large number of freshwater channels, the three largest of which empty into the Laptev Sea on average 65 %, 22 % and 5 % (Trofimovskaya, Bykovskaya and Olenekskaya channels respectively) of the total river

discharge (Magritskiy, 2001) (Fig. 1); the mean annual runoff volume of the river from 1935 to 2012 was about 539 km$^3$ (RosHydromet, web source). However, given the large territory of the Lena River basin and its outlet area in particular, direct measurements pertaining to the river are still insufficient. The high complexity of the region adds to the problem. As a result, the existing analyses of stream temperature and other discharge characteristics at the basin outlet are fragmentary and cannot provide the aggregate picture (Fofonova, 2014).

The goal of this paper is to analyse the available data on the water temperature of the Lena River at the basin outlet in the summer ice-free period (June–September). The analysis is based on long-term data series at Kusur (Kyusyur) hydrological station from the beginning of observations mainly to 2012, and additionally at several downstream hydrological stations and one upstream hydrological station (see Sect. 2.2.1 for details). These additional data are rarely used, but their analysis is critical for understanding the complexity of processes in the region. The analysis reveals the existence of a large

negative difference of the surface water temperature at Kusur gauging station (GS) and the beginning of the Bykovskaya channel at Habarova (Khabarova) GS during the open water season (from July to September) (Fig. 1). The description of this difference and factors that may be responsible for it is a particular focus of this paper.

In recent literature, the data on the Lena discharge and water temperatures at the Lena Basin outlet are, as a rule, taken at Kusur GS (e.g. Costard et al., 2007; Liu et al., 2005; Peterson et al., 2002; Yang et al., 2002; Yang et al., 2005),

situated ~200 km to the south of the delta head (Fig. 1). In this paper, we discuss to what extent the water temperature observations at this station represent the mean stream temperature. We show that the water temperatures measured at Kusur station fail to represent the mean cross-sectional value but reflect the thermal conditions of the Lena River in general. To verify this hypothesis and to explain mentioned difference the numerical experiments are set.

The paper is organized as follows. Section 2 presents a description of the data set used in this work, the hydrological

stations and measurement techniques. Section 3 contains analysis of water temperature tendencies at Kusur and Habarova stations. Section 4 presents a description of the surface temperature difference and its analysis. Section 5 contains description and results of the numerical experiments. In Section 6 and 7 we provide the discussion and conclusion respectively.

## 2 Description of hydrological stations, measurement techniques and the available data set

In this section we list the data available and used and the measurement techniques. We also describe the GS where

these data have been collected.



## 2.1 Measurement techniques and available data

Since the late 1930s, relevant data from hydrological observations in the Siberian region, such as discharge, water temperature, ice thickness, dates of ice events (ice cover formation and decay), are controlled and stored by the Russian Hydrometeorological Service. They are available in hydrological yearbooks in local centres of hydrometeorology and environmental monitoring and are partly available on the web (RosHydromet, web source). Table 1 lists the data available from the Russian Hydrometeorological Service used in this study. We also use CTD (conductivity-temperature-depth) data on water temperature profiles for several days in August 2011 at the cross-section of Habarova GS (Stolb, Bykovskaya channel) and Stolb, main channel, located 4.5 km upstream from Stolb Island. These data were collected during the Lena cruise of 2011 which was a Russian–German venture (Fofonova, 2014).

The Russian Hydrometeorological Service carries out measurements of water and air temperatures two times per day, at 8 a.m. and 8 p.m. Until 1993 in the USSR, the stream temperatures were measured at regional hydrologic stations on a 10-day basis (the 10th, 20th, and 30th days of each month) and were taken twice, at 8 a.m. and 8 p.m., on each observation day (State Hydrologic Institute, 1961). Measurements of the surface water temperatures covered from the end of spring, when the water temperature is close to zero, to the fall, a few days after the freezing of the water surface. The observations were made for flowing water; a cup with a thermometer was placed approximately 0.5 m below the water surface for five to eight minutes and retrieved carefully for a quick recording of temperature.

## 2.2 Description of the gauging stations

In this section we briefly describe the GSs referred to in this work (Fig. 1).

### 2.2.1 Kusur (70.70ºN/127.65ºE)

Kusur GS is located near Kusur Village at the site of the station carrying the same name (Fig. 1). The width of the stream there is 2.4 km on average for the summer season. The catchment area is about 2.43 million km$^2$. Measurements of stream surface temperatures are performed at the right bank of the Lena River on a distance ~3 m from a bank. The transverse profile of the riverbed in the area of Kusur GS is shown in Figure 2. Kusur GS has been operating since 1936 (Hydrological Yearbooks; RosHydromet, web source). At the moment the elevation of zero of gauge equals to −1.41 m (Baltic system of elevations). The water level varies in average from 16.5 m (in the beginning of June) to 7.8 m (late August) during the warm season (June–September) due to seasonal discharge variations (Fig. 3).

### 2.2.2 Habarova (Stolb, Bykovskaya channel, 72.42ºN/126.72ºE)

Habarova GS (Stolb, Bykovskaya channel) is situated in the area of the delta head at the beginning of the Bykovskaya channel (Fig. 1) on the territory of Stolb polar station, 7.7 km downstream from Stolb GS main channel. The width of the channel at the cross section of Habarova GS is up to 1.0 km. Measurements of stream surface temperatures



are performed on the right channel bank. Habarova GS has been operating since 1951 (Hydrological Yearbooks; RosHydromet, web source).

### 2.2.3 Tit-Ary (71.99ºN/127.09ºE)

Tit-Ary GS is situated on the right side of Tit-Ary Island, which consists of alluvial deposits. The river channel, with a width of about 12 km, is divided into two branches by the island. The island is 20 km in length, 7 km in width and 30 m in height and is located 1.2 km from the fairway. The left branch is shallow. Water temperature is measured on the right side of the island. The Tit-Ary GS operated for 15 years from 1976 till 1990 (Hydrological Yearbooks; RosHydromet, web source).

### 2.2.4 Eremeyka (70.41ºN/127.24ºE)

The Eremeyka River is a right inflow of the Lena River with a catchment area of 9.70 km$^2$. The station is located 2 km upstream from the mouth. Water temperature is measured at midstream. Eremeyka GS has been operating since 1974 (Hydrological Yearbooks; RosHydromet, web source).

### 3 Stream temperature characteristics at the basin outlet

In this section we focus on long term data for surface water temperatures at Kusur GS, which are usually taken as representative for whole basin outlet zone, and Habarova, situated in the delta head area, 200 km downstream from Kusur GS (Fig. 1). At the lower reaches of the Lena River (main channel, delta head area) the observations confirm that the water temperature vertical distribution is almost uniform for the entire ice free period (Reinberg, 1938) due to very high level of turbulent pulsations within the Lena River main stream and delta head area (Fig. 4). The Lena River discharge rate is ~42500 m$^3$ s$^{-1}$ on average from June to September (Fig. 3) and can reach 200000 m$^3$ s$^{-1}$ in the beginning of June or end of May. On average 22 % of total discharge passes through Bykovskaya channel, in the beginning of which Habarova GS is situated. The typical velocities during summer season are at about 1 m s$^{-1}$ (Hydrological Yearbooks) in the region from Kusur GS till Habarova, typical relative depth is 15 m within the main channel area (Figs. 1, 2) and is about 19 m midstream in the area of Habarova GS (mean depth is ~10 m, the riverbed profile is triangular shaped here). Only these estimations and typical water temperatures briefly give us Reynolds numbers (the ratio of product of mean flow velocity and mean depth to kinematic viscosity) higher than 1200·10$^4$. The available hydrological notes confirm that the vertical temperature distribution is uniform within cross-section at both Habarova GS and Kusur. Therefore, we assume that at the considered Lena River stations (Fig. 1) surface water temperature can be replaced by water temperature.

The fluctuations of mean monthly water temperatures in the surface layer usually follow the dynamics of mean air surface temperatures in the area closely (e.g. Johnson, 2003; Hammond and Pryce, 2007). A strong association between monthly stream temperatures at Kusur GS and monthly air temperatures in the Lena River basin outlet area has been shown by Liu et al. (2005). For August and September, their results are statistically significant at the 99 % confidence level. They



have also shown that the correlations between stream temperature and precipitation are very weak and statistically insignificant. We can confirm statistically significant correlations on a monthly scale of the water and air temperatures for both Kusur GS and Habarova (not shown, the correlation analysis was done between water and air temperatures for all summer months and September for all years) using available for us air temperature data and air temperature estimates given in the annual reports about climate characteristics in Russia provided by Institute of Global Climate and Ecology of the Federal Service for Hydrometeorology and Environmental Monitoring and the Russian Academy of Sciences (IGCE, web source; Table 1). In Table 2 the results of the recent water temperature estimates for both stations are summarized. Table presents the probability $p$ of null hypothesis 'no trend'. 'Plus' indicates that the $1 - p > 0.9$, which means presence of trend with level of statistical significance higher than 90 %. All trends found here are positive indicating the increase in the water temperature. The coefficients of the linear trends for the monthly averaged water temperatures are given (°C/10 years) in the brackets. Of course, the minimum level of statistical significance can be chosen higher or lower to determine the presence of trend, however, our goal is to show the overall dynamics.

Table 2 shows that the period from 1976 till 2011 is characterized by rapid water temperature growth. The same is valid for the air temperature within the Lena River watershed (IGCE, web source). If we consider period from 1951 to 2011 (Habarova GS has been operated since 1951) there is a tendency of the water temperature increasing during the early summer by 0.13 °C per decade at Kusur GS and Habarova. The estimations for the period 1976−2011 are different. If for the early summer there is a deceleration of the water temperature growth, the mid-summer is characterised by the acceleration of the growth. Also, the water temperature behaviour at Kusur GS and Habarova is slightly different during this time. The water temperature at Habarova GS demonstrates overall higher coefficients of the linear trends and higher level of statistical significance and has tendency to increase during the August. This is in agreement with air temperature changes. Sic coefficients of the linear trends for the air temperature averaged over warm season (May–September) show temperature increasing up to 0.8 °C per decade (in average 0.6) for the period 1976−2011 for the northern area of watershed and up to 0.6 °C per decade (in average 0.5) for the watershed area upper GS Kusur. Generally, air temperature is characterised by higher growth rate for the Lena River watershed area compared to the water temperature rate at the Lena River lower reaches. Also, the difference in the behaviour of stream temperatures at Habarova GS and Kusur indicates that the measurements at Kusur GS can be taken for analysis of water temperature changes in the delta head area with a great caution.

Figure 5a demonstrates mean water temperature over warm season (June–September) and maximum summer temperature for various years. It clearly shows that the mean water temperature at Habarova GS (10.79 °C) is higher than at Kusur GS (9.72 °C). This is not true for the maximum values, which, for example, close to each other quite often. For some years maximum at both station can reach 20 °C and higher. Figure 5b contains the information about time when the water temperature maximums are reached. For both considered stations the maximum water temperature is reached during July or first half of August (Fig. 5b). However, for Habarova GS there is a shift toward later date compared to Kusur GS. The calculated mean difference in the time, when the temperatures reach maximums, is about 2.3 days between Habarova



GS and Kusur for the period considered (1951–2011). This is in agreement with the mean time it takes the flow to reach Habarova GS starting from Kusur GS during summer. The correlation coefficient between maximum events at Habarova GS and Kusur is ~0.6 and significant at the $p<0.01$ level based on bootstrap analysis (see, e.g., in math.ntu, web source).

## 4 Water temperature measurements inconsistency and its analysis

5        The measurements inconsistency is detected during the analysis of water temperature *values* at the different stations, in particular Kusur GS and Habarova (Figs. 1, 5a). These inconsistencies cannot be easily identified from the analysis of monthly *fluctuation* of the water temperature during last 50–70 years. Typically, the water temperature in the Lena River gradually decreases toward its mouth in the summer months due to the river's south–north orientation (e.g. Liu et al., 2005; Zotin, 1947). In the lower reaches of the Lena River the presence of a deep valley and wide–open areas to the north and

northwest, together with being surrounded to the south by the Lena–Vilui lowlands, facilitates unhindered entry of cold air masses (Burdikina, 1961). However, the surface water temperatures measured at Habarova GS for all years of observations are on average much higher for the summer season than measured at Kusur GS (Figs. 5, 6) located much further upstream (Fig. 1). Figure 6 also clearly shows that the difference between water temperatures at Habarova GS and Kusur increases from June to September. The measured air temperature remains below water temperature for both stations.

Below, we discuss the possible causes of this large positive difference between water temperatures at Habarova GS and Kusur.

a) The anthropogenic factor as a possible explanation should be discarded immediately given the very low population density in the region and the absence of industrial facilities and dams.

b) The features of the river–atmosphere heat exchange could be a possible missing factor. The summer period is

characterized by a strong short-wave radiative forcing, especially until 7 August during the polar day period (Langer et al., 2011). Despite the lower air temperature the water temperature still can increasing due to large portion of the short wave radiation. Figure 7 shows the daily averaged values of the heat balance components for the period from 2002 till 2011. The albedo of the Lena River water was set to 0.1. The sensible heat fluxes were calculated using Edinger et al. (1974) formula. The wind speed, humidity and air temperature were taken from observations at Kusur meteorological

station (provided by Arctic and Antarctic Research Institute). The shortwave and longwave incoming radiations were taken from the National Oceanic and Atmospheric Administration database (NCEP/NCAR Reanalysis, web source). The net radiation balance at the GS Kusur and Habarova is positive during period from June till mid-August and negative during September. For some years net radiation balance becomes negative in the middle of August or even in the beginning of August. In the area of Habarova GS cooling starts earlier due to smaller air temperature (Figs. 6, 7).

However, the net heat balance (sum of net shortwave and net longwave radiation heat fluxes, sensible and latent heat fluxes) logically tends to decrease from June to September within the studied area, what is nearly opposite to behavior of difference between water temperatures at Habarova GS and Kusur. The two important factors, the heat balance and heat accumulated upstream from Kusur GS, which should largely explain the mean monthly stream temperature values, fail to



do so at Habarova GS.

c) Ice conditions and as a result additional latent heat fluxes with large magnitude can also be a factor of influence. However, during July–September in the area of interest there is no floating ice.

e) The heat exchange with a river bed is still missing in our reasoning. However, it is obvious that during July − August the heat should be transferred from water to the sediments, what can be seen even for the lakes in the area, which have smaller heat content (Boike et al., 2015). In the fall the back heat fluxes from sediments to river water should be expected. This fact also complicates the picture, because in the area we can expect additional factor, which tends to cool the water during the summer. In work by Boike et al. (2015) for the lakes situated on Samoylov Island, the temperature in which reaches 15 °C, the heat fluxes from the water to the sediment do not exceeded 3 W m$^{-2}$ for the summer season. However, there is a large gap in understanding of the processes of the water–river bed heat exchange in the permafrost area. Some details about this issue will be presented in the next sections.

d) The possible reason for this puzzling disagreement could be the non-representativeness of measurements at one or both the stations. We should stress that water temperature measurements at both station are taken near the right riverbank. The stream temperature measured near the bank does not always correspond to the true mean stream temperature. This highly depends on local conditions like inflows with different temperatures upstream, the shallowness of the water layer or other coastal effects. On the other hand, the Lena River within the main channel has very strong vertical and lateral mixing during summer season  (Fig. 4).

According to the results of temperature surveys in 1979 and 1985 provided by the hydrometeorology and environmental monitoring centre in Tiksi, the temperature measurements at Habarova GS are representative. The absolute differences between surface temperatures near the bank and midstream did not exceed 0.2 °C.

However, several hydrological notes from 1930s, 1950s and 1980s (provided by centres of hydrometeorology and environmental monitoring in Tiksi) mention the possibility that surface water temperature measurements at Kusur GS lack representativeness. The differences between the weighted average and near coast stream temperatures ranged from 1 to 3.5 degrees and always remained positive (the measurements have been done in July and August). Based on observations in 1936, the mean ratio of these temperatures was found to be 1.2 for the warm season (June−September) (Reinberg, 1938; Zotin, 1947). Taking into account the technique of measurements and the river bed profile (Fig. 2), which shows a sharp increase in depth near the shore, we can assume that the main reason for non-representativeness is the influence of relatively cold water from several small inflows represented by Tikian, Bordugas, Abadachan, Ebitiem (Ebetem) and Eremeyka Rivers. The mouths of these rivers are located approximately between 20 km and 1.5 km upstream from Kusur GS on the same river side (Balashov and Tamarskiy, 1938). In the whole area of interest till Habarova GS there are no other inflows, which could affect the temperature measurements at the stations considered. The main question, which arises here, is whether there is a possibility of cold right bank current formation, which persists till Kusur GS.  Does the cold water from these rivers mix not fully with the relatively warm water of the Lena River till Kusur GS? The mean annual volumes of the Ebitiem and Eremeyka runoffs are only 0.4 and 0.0034 km$^3$ respectively





(these estimates are provided by centres of hydrometeorology and environmental monitoring in St. Petersburg and Tiksi), for other small rivers we can only guess that it is about 0.2 km$^3$. Therefore, the water from Ebitiem River, which mouth is ~5 km upstream from Kusur GS, dominates the cold current formation. Unfortunately, we do not have temperature data for the Ebitiem River, but the daily course of the Eremeyka water temperature near the mouth averaged over the period from 2002 to 2001 is shown in Figure 6.

To find out the influence of water from the small rivers mentioned above on water temperature measurements at Kusur GS and to carefully estimate the water–air heat exchange we made several computations, which will be presented in the next section.

## 5 Numerical experiments

We made two numerical experiments using COMSOL Multiphysics, in particular, Computational Fluid Dynamics Module (Wilkes, 2002; COMSOL, web source). K-epsilon (k-ε) turbulence model was used to parameterize both horizontal and vertical mixing. The wall functions are used to describe the flow motion near the riverbed. The roughness height  was set to 3.2 m and roughness parameter to 0.26, which corresponds to a sandy, loam soil. The main purpose of the first experiment is proving the hypothesis that very small tributaries upstream Kusur GS can influence the measurements taken near the right river bank and getting some quantitative characteristics of the influence. The second experiment has been designed to reproduce the temperature at Habarova GS using atmospheric forcing and results from the first experiment.

1)      First experiment. The model domain was constructed as a box with a length, width and depth equal to 20000 m, 2400 m and 15 m respectively. The rectangular grid was generated with a resolution 100 m, 10 m and 1 m in along channel, cross-sectional and vertical directions respectively. The Lena River discharge was set to 25000 m$^3$ s$^{-1}$ , which  corresponds to the typical water velocities of about 1 m s$^{-1}$ for 15 m depth. The atmospheric forcing was turned off. The water temperature in the tributaries was taken equal to Eremeyka water temperature at the appropriate time. Discharge rate for the Eremeyka River was available from observations, however, only on monthly scale (Table 1). The discharge rates for the other tributaries were calculated approximately based on the available information about the watershed square and shape of the channels and were scaled according to the behavior of the Eremeyka discharge. Total discharge from all tributaries was varying in a range from 300 (beginning of June) to 17 m$^3$ s$^{-1}$ (end of August), the averaged values over the period from 2002 to 2011 were set to 132 m$^3$ s$^{-1}$ for June, 75 m$^3$ s$^{-1}$ for July, 61 m$^3$ s$^{-1}$ for August and 80.5 m$^3$ s$^{-1}$ for September.

Numerical simulations showed the possibility of a thin layer formation, at about 170 m from the right river bank to midstream, of the relatively cold water due to the influence of  tributaries. Note that the width of the channel is 2 orders of magnitude larger than depth, and that in our idealized experiment the difference between surface water temperature and bottom temperature did not exceed 0.2 °C .

Varying the different turbulence schemes and discharge conditions of both Lena River and its inflows the width of the layer, which is experiencing the impact of the small cold tributaries, remains nearly constant. The width of this layer depends on the characteristics of the cross-sectional turbulent boundary layer formation and on turbulent heat transfer coefficient. There





is a nearly linear dependency between the elevation and discharge at Kusur GS during July–September due to a nearly rectangular profile of the channel (Fig. 1). Thus, typical velocities over different discharge conditions and full cross-sectional width at Kusur GS vary slightly. In our simulation we neglected variation of the water viscosity (Reynolds numbers are higher than $1200 \cdot 10^4$) and turbulent heat transfer coefficient; as a result, the width of the mentioned layer varied negligibly small. In an idealized case with a plate equipped with heat sources the temperature distribution in the turbulent boundary layer follows the logarithmic low except for the thin wall layer for the flows with very high Reynolds numbers (Landau and Lifshitz, 1987). In our case, setting the same temperature for all inflows, we obtained a close to logarithmic profile of the water temperature distribution horizontally within the layer of 170 m width. It is highly expected due to the use of wall functions. Assuming that the inflow velocity of tributaries is negligibly small, we can describe the behaviour of midstream water temperature using the following approximation:

$$\frac{1}{(L-m)} \int_{m}^{L} f(x)\, dx = a \cdot T_e + b \cdot T_l, \tag{1}$$

$$a + b = 1, \quad a = \frac{Q_e}{Q_e + Q_l * \dfrac{L}{L_{cs}}} = \frac{1}{1 + \dfrac{Q_l}{Q_e} * \dfrac{L}{L_{cs}}}, \tag{2}$$

$$f(x) = \frac{T_l - T_k}{\ln\left(\dfrac{L}{m}\right)} \cdot \ln\left(\frac{x}{m}\right) + T_k. \tag{3}$$

In these formulas $T_e, Q_e$ are the Eremeyka water temperature and total discharge rate from all small cold tributaries upstream, $T_l, Q_l$ are the Lena water temperature and discharge rate, $L$ is the width of the layer, where the influence of cold water from tributaries takes place, $L_{cs}$ is the width of whole cross-section at the Kusur GS, $m$ is the distance to the right river bank, at which the measurements of water temperature were taken (we set it to 3 m), $f(x)$ is a function of temperature distribution, which depends on distance $x$ to the right Lena River bank.

Using Eqs. (1)–(3) the midstream water temperature, which is close to mean stream temperature $\left(\frac{L}{L_{cs}} > 10\right)$, can be written as:

$$T_l = \frac{d \cdot T_k - a \cdot T_e}{b - c}, \tag{4}$$

$$c + d = 1, \quad c = \frac{1}{1 - \dfrac{m}{L}} - \frac{1}{\ln\left(\dfrac{L}{m}\right)}. \tag{5}$$

Using Eq. (1), the mean, maximum and minimum difference between the midstream and near right bank temperature were calculated (Figs. 8, 9).





Figure 8 demonstrates that the influence of cold tributaries increases from June to the beginning of September in general. The mid-stream temperature is on average higher by 0.8 °C than the near bank temperature during July − September. It means that the cold tributaries can explain, at least partly, the large positive difference between the temperatures measured at Habarova GS and Kusur. The influence of the tributaries can be causing warming but mainly in June and only for some

particular years. In Figure 9 negative values of minimum of the difference between the midstream and near right bank water temperatures in the middle/end of June correspond to the higher temperature in the Eremeyka than in the Lena River in June, 2004. Figure 9 illustrates that the cooling influence of inflows can greatly vary and its magnitude can reach 5.5 °C under certain conditions. One of the strongest factor determining the influence is the ratio of discharge rates of the Lena River and its tributaries (Eqs. (2,4)). In current simulations we used monthly values of discharge for the tributaries (kept discharge at

the same level for the whole month), this explains why all maximums for the particular month (July, August, September) are attributed to one particular year (Fig. 9). So, in 2003 the tributaries have anomalously high mean September discharge rate, in 2007 the August discharge was higher than usual. However, the discharge rate influence can be enhanced or weakened by the water temperature in the tributaries. For example, in 2011 the discharge rate in July was smaller for tributaries than in 2003 and quite similar to 2003 for the Lena, however, the temperature difference in 2011 is much higher than in 2003

(Fig. 9). If the water temperature in Eremeyka and other tributaries is much colder than in the Lena River, then the non-representativeness of the measurements becomes more pronounced. At the end of August and beginning of September both factors are usually working, the discharge rate of the Lena River is decreasing (Fig. 3), the temperature is increasing compared to that of tributaries, that is why the curve of mean influence tends to increase from June till the beginning of September. In June (especially in the beginning) the influence of the cold tributaries usually nearly vanishes due to the large

Lena River discharge rate (Fig. 3) and small temperature gradients.

The water temperature characteristics modeled for the years 2003, 2007 and 2011 correspond to the large difference between Kusur and Habarova GS water temperatures for particular months. Unfortunately, we do not have daily values of the discharge rates and temperatures for all tributaries (daily water temperatures and monthly discharges are available only for Eremeyka), which are very important to determine actual values of midstream Lena water temperature for particular dates.

The curves presented for different years (Fig. 9) do not reflect daily behavior of the difference between the midstream and near right bank water temperatures realistically, because the discharge rates usually significantly vary during one month and it is hard to speculate about the typical seasonal curve of the discharge for tributaries. The above estimates for the midstream Lena water temperature present a useful benchmark, but contain a lot of uncertainties. For example, in our idealized experiment we did not turn on the atmospheric forcing, which can be a significant source of the surface stress and can both

reduce the non-representativeness of the measurements or enhance it. If we add large wind stress to the system, then $L$ and shape of $f(x)$ cannot be considered as fixed in time anymore. However, winds with speeds 5–6 m s$^{-1}$ were prevailing during the considered period of time, the winds with magnitude larger than 18 m s$^{-1}$ were not present.

Proving the influence of the small cold tributaries on the measurements at GS Kusur, we should discuss the justice of the results and estimates, which were given above for the near bank water temperature, for the mean cross-sectional water





temperature. The midstream temperature is systematically higher than the measured at the river bank on a monthly scale for the period from July to September. However, we can estimate now the role of the Lena and Eremeyka water temperature in formation of the Kusur temperature (Eqs. (2,4) and (5)). The mean Lena contribution is 90 %, 88 % and 85 % in July, August and September accordingly. The water temperature in tributaries is also affected by the regional atmospheric forcing.

The correlation coefficient between monthly water temperature measured at Habarova GS and Eremeyka is ~0.86 (the data set of 148 points contains monthly mean values for open water season from 1974 to 2010). Thus, we can assume that the trend and mean heat balance estimates at the Kusur GS can be taken for the Lena River midstream with caution, but the non-systematic component of the difference between midstream and right river bank temperatures adds additional noise, which reduces the accuracy of the assessments. The mean net heat flux will be a bit smaller for the Lena River midstream compared

to the one presented in Figure 7 for July−September by about −10−20 W m$^{-2}$ due to a higher gradient between the water and air temperatures. The estimations with higher accuracy require knowledge of daily discharge rates and temperatures for all tributaries closely upstream Kusur GS.

2)    For the second experiment we took a segment from Kusur GS till Habarova and turned on the atmospheric forcing. Here we should mention that the Lena water is highly turbid. According to the observations in June–July (Örek et. al, 2013)

the light penetration depth (Secchi disc depth) was in the range 30–90 cm at the Delta head area. We used this information to estimate the penetration depth of shortwave radiation. Year 2012 was chosen as a modelling year because additional information about the elevation for Eremeyka River was available (Table 1). Note that the elevation measurements at Eremeyka are not influenced by Lena because the elevation of zero of Eremeyka GS (36.28 m) is higher than the possible Lena water level. The atmospheric forcing was derived from the NCEP/NCAR Reanalysis, web source. We used daily data

for the discharge and water temperature at Kusur GS, daily data for the water temperature in Eremeyka to set up and force the experiment. Additionally, to identify the influence of the small tributaries upstream Kusur, the optimization task was posed for the total daily discharge rate from all tributaries, which is unknown. The difference between the modeled and measured water temperatures at Habarova GS was minimized using 40 points equally distributed along the time line (June– September). Figure 10, top panel, demonstrates the total discharge from all small tributaries within the warm season of 2012,

which is the result of optimization task. Independently, from previous estimates of the total discharge we obtained nearly the same range, however, with small mean value at about 41 m$^3$ s$^{-1}$. This is in agreement with the fact that in 2012 the mean summer discharge rate of Eremeyka was smaller than usual. In Figure 10, top panel, the mean monthly discharge rate of Eremeyka River multiplied by 400 is also presented. Note that the Eremeyka water doesn't play the major role in the formation of cold right river bank current. The discharge rate of Ebitiem River (5 km upstrem Kusur GS) is more than

100 times larger than the rate of Eremeyka on average. It can be seen that the mean monthly discharge rates and elevation at the Eremeyka River are in agreement with the optimized daily discharge rates during summer season, except for June. However, as mentioned before, in June the floating ice can be present, which would modify the water heat balance a lot. Figure 10a (bottom panel) shows simulated and measured temperatures at Habarova GS and demonstrates that they agree quite well, with mean error 0.4 °C. Comparing Figure 10a (bottom panel) and 10b (bottom panel) we can conclude that the





warming influence of the atmosphere within the area studied (~200 km) due to large short wave radiation heat fluxes in June/beginning of July is limited to 0.5 °C (can reach 1.5 °C), in the end of July–August the warming effect adds about 0.2 °C to Habarova water temperature and then weakly expressed heating is gradually replaced by cooling. Figure 10b (bottom panel) demonstrates the findings of previous experiments for 2012. The midstream temperature (close to mean

stream value) at Kusur GS can be significantly higher than the right river bank temperature for some dates, up to 4 °C in the beginning of August for 2012, due to cooling influence of small tributaries upstream.

        However, for some years we cannot explain large temperature difference at Habarova GS and Kusur, which can be up to 8 °C, even solving optimization tasks (we varied only total discharge from tributaries, its water temperature is taken as it is), this occurs for some years in the beginning/middle of June (2009, 2011) and in the beginning/middle of September

(2007). Due to missing information about ice conditions in June and its possible large influence on the water temperature measurements, we focus our attention only on September mismatch. Figure 11, top panel, shows the optimal total discharge rate from all tributaries and mean monthly discharge rate of Eremeyka River multiplied by 400 for 2007. The obtained range of discharge rate from 10 to 290 $m^3\ s^{-1}$ agrees with the estimations presented above (no upper and bottom limits were introduced for the total tributaries discharge rate during the optimization process). The difference between the modeled and

measured temperatures is reasonable before September except for June (Fig. 11), the mismatch between the modeled and measured temperature in the middle of July can be removed introducing larger frequency of optimization process. However, in the end beginning/middle of September the difference between the modeled and measured temperatures at Habarova GS reaches ~6 °C. The inflows during this period of time have warming effect, thus a sharp drop in optimized discharge can be seen (Fig. 11). Note that the atmospheric forcing tends to rapidly cool the water from Kusur GS to Habarova in the middle of

September.

        There is an indication in favour of an unaccounted source of heat in the middle of September 2007 from the riverbed in the area of the delta head. More analysis and observations are required to make further statements in this direction. Some considerations are presented in the next section.

## 6 Discussion

One of the questions, which is still open, is the influence of stream bed on the water temperature within the area from Kusur GS till Habarova. The arctic location of the Lena Delta secures its position within the continuous permafrost. Frozen ground thickness in the region can reach 600 m (Grigoriev, 1966), active layer thickness is rarely exceeding 0.8 to 1.2 m. Taliks usually occur below the large water bodies, such as lakes and river channels; talik zones are mostly 'open' beneath the major channels and largest lakes, while remaining ones under the secondary branches and smaller water bodies

are 'closed'. The presence of an 'open' talik under the Lena River main channel within the studied area is very likely (Grigoriev, 1993), taking into account that the Lena River does not freeze completely till the bottom during the winter season (Hydrological Yearbooks). It means we cannot expect much larger heat fluxes from the river to river bottom than estimated in Boike et al. (2015) for lakes, however, the processes of the stream–subsurface exchange within the hyporheic zone leave



a question and can be an important factor. Here we should stress that the width of the main channel in the area studied is two orders of magnitude larger than the depth, the level of the river water within the area studied varies significantly during summer season, it means that the heat exchange of the river with river banks is limited compared to the heat exchange with the river bottom.

5        Due to a very high Lena River summer discharge the constant heat fluxes from atmosphere to the water at about 200 W m$^{-2}$ in June induce only less than one degree water temperature rise within 200 km. It means that the noticeable change in the river water temperature due to the influence of river bed, if it is, should be searched in a shallow part with a tranquil flow. Within the area studied, the river is tighten closely by Kharaulakh Ridge on the east and by Czekanowski Ridge on the west, that is why the Kusur bed profile is nearly rectangular (Fig. 2); some sandbanks can be found within right

Lena River bank. Additionally, there is a large island Tit-Ari (20 km in length, 7 km in width and 30 m height) within the area studied, located between Kusur GS and Habarova, which is a remnant of the high floodplain probably dating back to the latest stage of the Flandrian transgression and which preserves a northernmost larch forest colony in the region and rich variety of mosses (Ivanova et al., 2012). It is known that the presence of an organic layer restricts the ground thaw due to lower thermal conductivity compared to mineral soil (Woo and Rouse, 2008). The large portion of the latent heat is needed

for the active layer creation. At the Tit-Ary Island, the thickness of active layer in August varies in a range from 15 cm to 1 m (Ivanova et al., 2012). Severe temperature conditions favour the closing of seasonal frost with a permafrost table with a very low temperature −10 −13 °C (two–sided freezing). Soil freezing is accompanied by frost cracking, cryoturbation and heaving of soil material. A combination of organic and mineral soils produces distinct flow mechanism on the Tit-Ary Island, there is developed drainage channel networks with a system of lakes, some of them are situated in the south–western

part of the island. All these facts favor that Tit-Ari Island can accumulate large amount of very cold water, and represent the large source of latent and sensible heat fluxes which tends to cool the water during warm season and can be higher during July–September than the net positive influence of the atmosphere. On the eastern part of the island the Tit-Ari GS (Fig. 1) was operated from 1976 to 1990. The river bed profile in the area of Tit-Ari GS shows a very gently slope presented by sands near the right bank of the Tit-Ary Island. The measurements of the river water temperature were taken in the close

proximity to the island, it means in a very shallow part (~2m). Table 2 shows that the temperature at Tit-Ary GS is typically smaller that measured at Habarova GS and Kusur in July–August. The hydrological notes dated before 1976 confirmed this behaviour. For some years the measured water temperature at Tit-Ary GS can be a little bit higher (not more than 0.8 °C) than that measured at Kusur GS.

        Unfortunately we do not have information about the representativeness of these measurements for the whole cross-

sectional area. Assuming the representativeness of measurements at Tit-Ary GS, it is becomes impossible to explain the water temperature at Habarova GS without introducing additional large positive heat flux from the river bed in the area of GS Habarova for the period from July to September and negative heat flux from the river bed in the area from Kusur to Tit-Ary (Fig. 1). We assume that the measurements at GS Tit-Ary are not representative for the whole cross-section due to





the influence of the Tit-Ary Island and its accumulated water. We can conclude that the large cooling influence of the river banks can be find within the area studied, but most likely it has a localized character.

However, we are forced to look on a possible warming effect of the riverbed (Fig. 11). Also the beginning of ice conditions at Habarova GS is observed on average four days later than at Kusur GS based on available observations from 1986 to 1990 and from 1999 to 2007 (Table 3). Ice formation is a complex process, but it largely depends on heat exchange with the atmosphere and heat stored in the river (Antonov, 1961). The date of fall ice appearance is taken as the date of formation of stable slush ice run (shuga drift) and drift ice (in this sense, the presence of small inflows upstream GS Kusur should have minor influence on observations). Despite the difficulty in determining this date, Kusur GS is considered to be one of the most representative for surveillance regarding ice phenomena (Antonov, 1961). Given that the air temperatures are nearly equal at Kusur GS and Habarova for the first decade of October and cooling influence of the atmosphere within the area studied, we conclude that the shift in the beginning of ice conditions is mostly explained by the impact of heat stored in the sediments. The accumulative environment of the Lena Delta significantly limits sediment delivery to the marine zone. Following the inter-annual variability of the river flow, the annual suspended sediment load (SSL) varies from 16.6 to 26.2 mln t, as measured at Kusur GS (Korotaev, 2012; Holmes et al., 2002; Hasholt et al., 2005). The vast majority of SSL passes by the Kusur cross-section in early summer when snowmelt events provide around 85 % of the total water discharge. Suspended sediment concentrations, on average, peak later than does the discharge, reflecting the dominant role of more distant material sources and the erosion–limiting setting of the Lena lower reaches (Tananaev, 2013). According to the results presented in Tananaev and Anisimova (2013) and Alekseevskiy (2004), annual bedload flux at Kusur GS is 14.9 mln t, which comprises nearly 42 % of the total sediment delivery to the delta head. Bed material transport occurs mostly during snowmelt floods (78.5 %). This is followed by rain–induced events (19.5 %) and the summer low flow period (2 %) (Tananaev and Anisimova, 2013). The vast majority of sediment material is retained within the riverine part of the delta. Presumably, the whole volume of bedload material is retained within the delta in large bedforms especially in the delta head area. Only 10 to 17 % (2.1 to 3.5 mln t) of the total suspended material is delivered to the Laptev Sea margin (Peregovich et al., 1999; Rachold et al., 1996). Sediment associated heat flux is expected to have higher impact within the deposition area, which includes delta head area and beginning of Bykovskaya channel, where GS Habarova is situated. Based on the results of the expedition in August 1955, 1959 in the Bykovskaya channel, no frozen soils in the furrows have been found, bed deposits were composed by sands, pebbles and boulders through entire depth of observations, which was ~8.5m (Ivanov, 1967). We can conclude that the heat fluxes from sediments to Lena water in the delta head area is larger than these in lakes estimated by Boike et al. 2015, to have precise estimations additional observational data are needed. However, the picture, which we obtained for 2007 (Fig. 11), remains very questionable. Even if we suppose that sediment strata actively starts losing its heat in the beginning of September, we cannot explain such warming without introducing additional large positive heat flux from the hyporheic zone in the delta head area. There is an evidence for the presence of a variety of cavities and groundwater flow systems on talik under the main channel. Exactly in August 2007 wedge cavity was



detected, which was closed by sands in 2008 and 2009, in the delta head area (Fig. 12), this fact opens more questions about dynamics in the system river water–riverbed and indicates necessity of future investigation in this direction.

**7 Conclusion**

This paper analyses water temperature characteristics in the outlet area of the Lena River during the summer season (June–September). Based on our analysis, we conclude that the measured water temperature at Kusur GS close to right river bank does not represent the mean stream temperature, underestimating it in July–September. The water from small Lena River tributaries (Eremeyka, Ebitiem, Beris and others) 1.5−20 km upstream GS Kusur forms relatively cold right bank current (except for June for some years, when the formed current can be warmer than Lena water), which influences the measurements. The ratios of the discharge rates of the Lena River and small inflows upstream and water temperature gradient of inflows and Lena River are the major factors which control the difference between the midstream (close to mean stream) and near right bank temperature, which is usually largest in the end of August, beginning of September. The midstream temperature is in average higher by 0.8 °C than the near bank temperature during July–September. However, the cooling influence of inflows can greatly vary and its magnitude can reach 5.5 °C under certain conditions. To recover the midstream temperature reliably the information about discharge and temperature conditions in the inflows should be collected.

At both Kusur GS and Habarova GS there is a tendency toward increasing water temperature. The estimates varies in a limit 0.07−0.25 °C per 10 years for different months and different stations (Table 2). The difference in the behaviour of stream temperatures at Habarova GS and Kusur and non-representativeness of the measurements at Kusur GS for the whole cross-section indicate that the measurements at Kusur GS should be taken for analysis of water temperature changes in the delta head area with a great caution.

There are indications in favour of an unaccounted source of heat in the late summer/beginning of fall from the riverbed to the water in the area of the delta head. More analysis and observations are required to make further statements in this direction.



## Author contribution

V. Fofonova, M. Krayneva and D. Yakshina were analysing available long term data. V. Fofonova and I. Zhilyaev designed and carried out the numerical experiments. N. Tananaev was consulting team about morphology of the Lena River bed and sediment processes in the region of interest and wrote appropriate text parts. N. Volkova and V. Fofonova were collecting long term data from printed materials and organized them in digital tables. K.H. Wiltshire provided CTD data, supervised team and organized trips to St. Peterberg and Tiksi.

## Acknowledgments

We are indebted to L. Ivanova, V. Natiaganchuk, A. Kraberg, O. Semenova, I. Fedorova, D. Bolshiyanov, V. Ivanov and A. Makshtas for their invaluable assistance in finding data and scientific consulting. We express gratitude to local centres of hydrometeorology and environmental monitoring for providing these data. We also thank S. Danilov and E. Golubeva for their valuable comments and German Federal Ministry of Education and Research (BMBF) for financial support, project "LenaDNM", grant identifier is 01DJ14007.

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





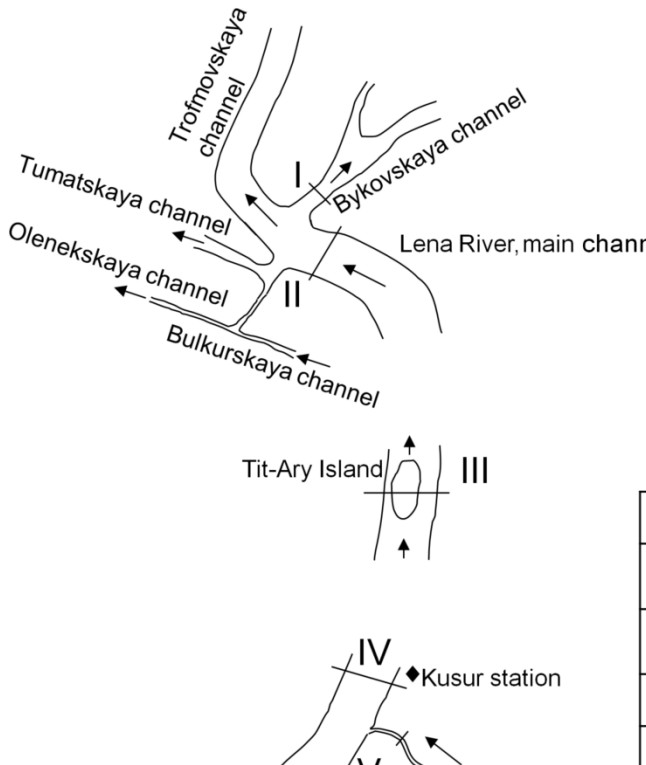

I – GS Habarova (Stolb, Bykovskaya channel)

II – GS Stolb, main channel

III – GS Tit-Ary

IV – GS Kusur

V – GS Eremeyka

| The objects on the map | Distance |
|---|---|
| Mouth of Ebitiem River – GS Kusur | ~5 km |
| Mouth of Eremeyka River – GS Kusur | ~1.5 km |
| GS Kusur – GS Habarova | ~200km |
| GS Tit-Ary – GS Habarova | ~50km |

**Figure 1:** The scheme of gauging station (GS) locations.





**Table 1:** The time resolution of available data for the warm season, which are used in current work.

| Station | Data type | | | | | | | |
| --- | --- | --- | --- | --- | --- | --- | --- | --- |
| | Surface water temperature | Surface air temperature | Wind conditions | Date of maximum daily water temperature within the year | First ice appearance date in fall | Humidity | Discharge rate | Elevation |
| **Kusur** | daily 2002–2012, 10 days 1936–2011 | daily 2002–2011, monthly 1978–2010 | 3 hours 2002−2011 | 1936–2012 | 1986–1999, 2000–2007 | daily 2002−2011 | daily 1936−2008, 2012, monthly 1935–2011 | daily 2002−2012 |
| **Habarova** | daily 2002–2012, 10 days 1951–2011 | daily 2002–2011 | – | 1951–2012 | 1986−1999, 2000−2007 | – | – | – |
| **Eremeyka** | daily 2002–2012 10 days 1974–2011 | daily 2002–2011 | – | 2002–2012 | – | – | monthly 1974–2012 | daily 2012 |
| **Tit–Ary** | monthly 1981–1990 | – | – | – | – | – | – | – |

| Lena River watershed | Linear trend coefficients for the surface air temperature | | | | Deviation from the mean air temperature value for the period 1961–1990 | | | |
| --- | --- | --- | --- | --- | --- | --- | --- | --- |
| | seasonal 1976–2011 | | | | annual 1936–2011 | | | |


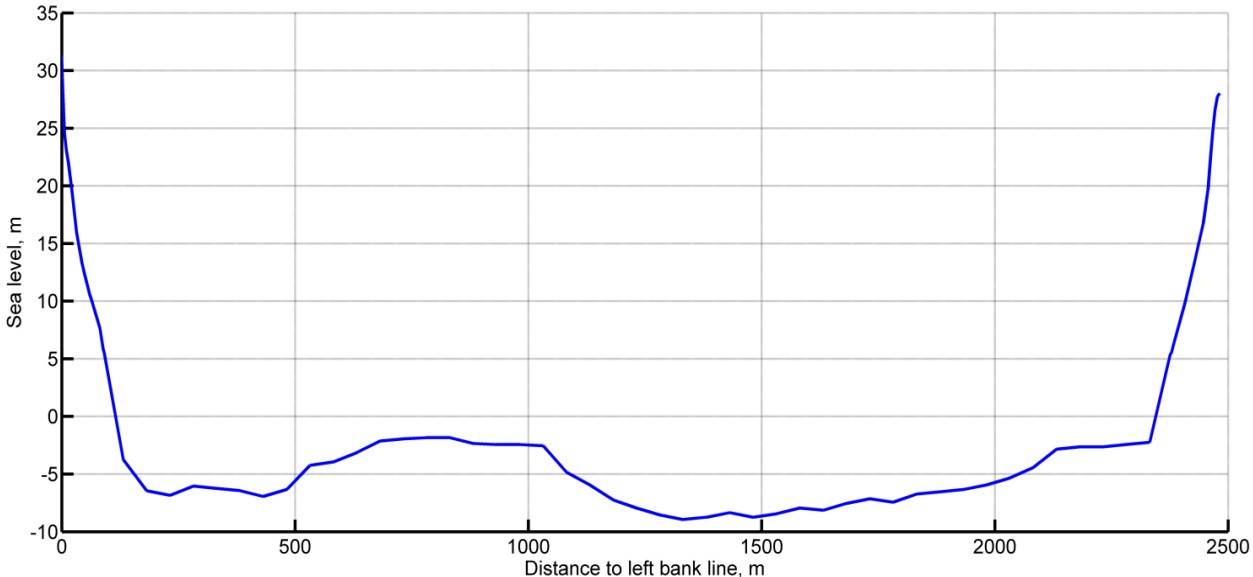

**Figure 2:** The transverse profile of the riverbed in the area of Kusur GS based on observations in 2012, first decade of June, m.

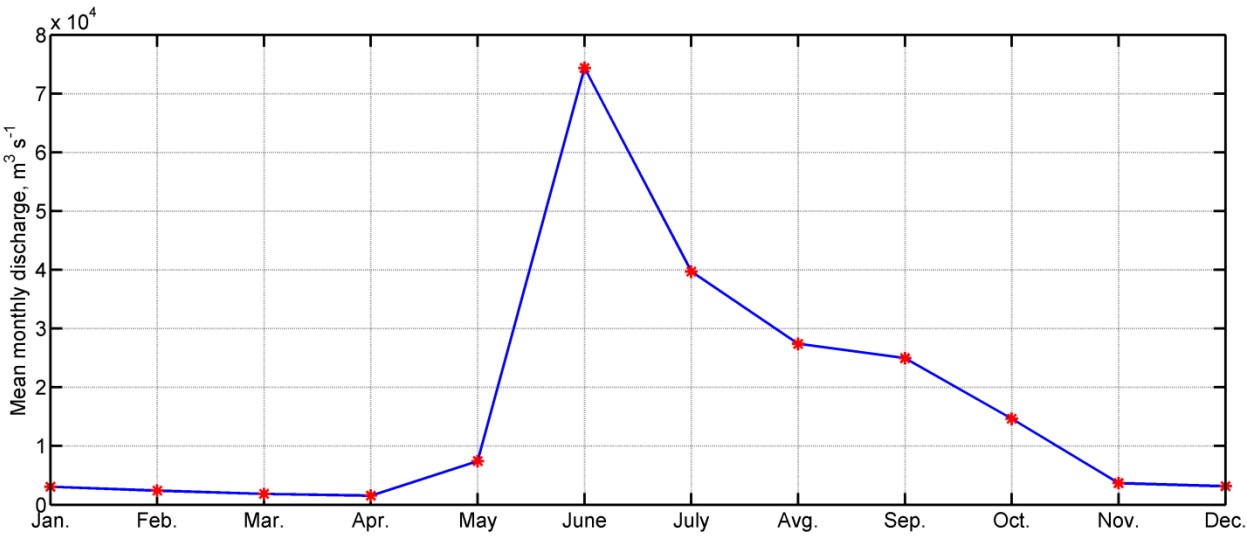

**Figure 3:** The mean monthly discharge for the period from 1935 to 2011, m$^3$ s$^{-1}$.




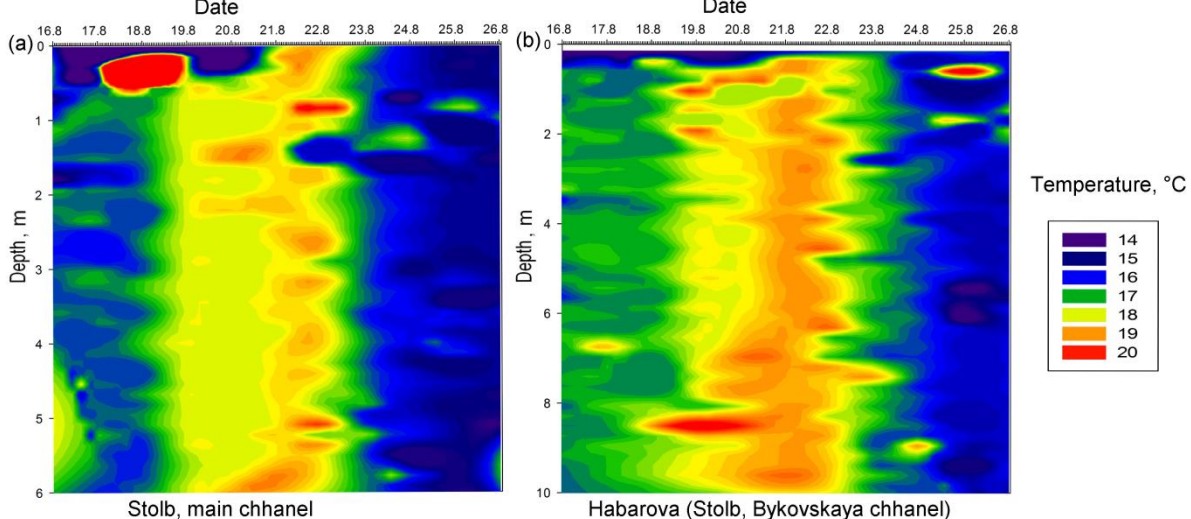

**Figure 4:** Stream temperature profiles from CTD measurements, which were taken in August, 2011. Depth is counted down from the free surface.



**Table 2:** Water temperature estimates for the different periods of time: (a) at Kusur GS; (b) at Habarova GS.

| (a) Kusur GS | Period | | |
|---|---|---|---|
| | 1936–2011 | 1951–2011 | 1976–2011 |
| Month(s) | Probability of '0' hypothesis (= 'no trend'), $p$, '+' indicates $p < 0.1$ ((1-$p$)·100 % > 90 %) and is followed by trend assessment | | |
| June | 0.322 | 0.1729 | 0.3552 |
| July | 0.222 | 0.0757, + 0.13 °C/10 years | 0.02049, + 0.23 °C/10 years |
| August | 0.0497, +0.13 °C/10 years | 0.1692 | 0.582 |
| September | 0.7573 | 0.9143 | 0.941 |
| June–September | 0.19 | 0.0832, + 0.08 °C/10 years | 0.0981, + 0.1 °C/10 years |

| (b) Habarova GS | Period | |
|---|---|---|
| | 1951–2011 | 1976–2011 |
| Month(s) | Probability of '0' hypothesis (= 'no trend'), $p$, '+' indicates $p < 0.1$ ((1-$p$)·100 % > 90 %) and is followed by trend assessment | |
| June | 0.07287, +0.13 °C/10 years | 0.2613 |
| July | 0.1164 | 0.00407, + 0.25 °C/10 years |
| August | 0.4704 | 0.05793, + 0.16 °C/10 years |
| September | 0.8189 | 0.1707 |
| June–September | 0.07038, +0.07 °C/10 years | 0.00498, + 0.16 °C/10 years |




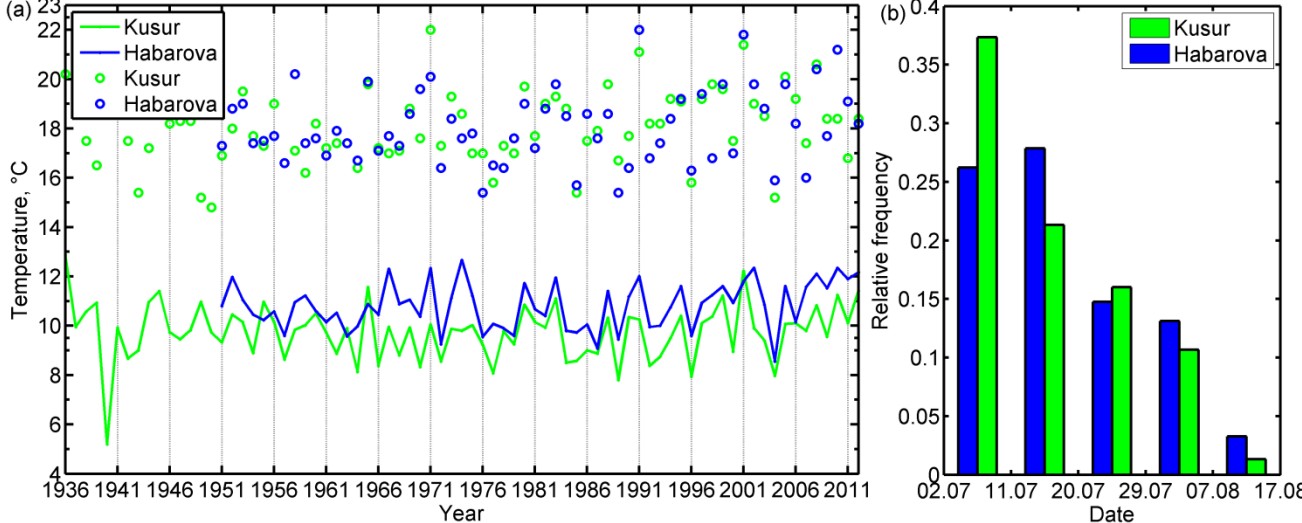

**Figure 5:** (a) The mean summer temperature (June–September) at Kusur GS and Habarova for the years from 1951 till 2012, °C. Dots indicate the maximum summer temperatures at both stations. (b) The relative frequency indicating how often the maximum is reached during the current period of time at both stations considered.

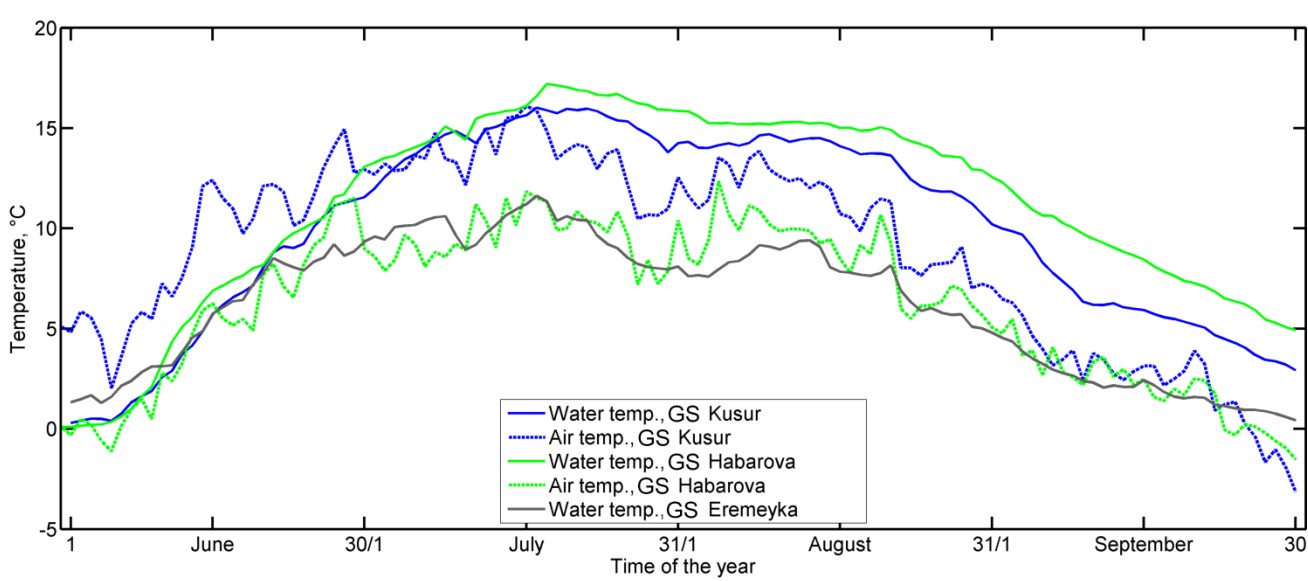

**Figure 6:** The mean daily surface air (2 m) and water temperatures measured at Kusur GS, Habarova and Eremeyka for the summer season (2002–2011), °C.





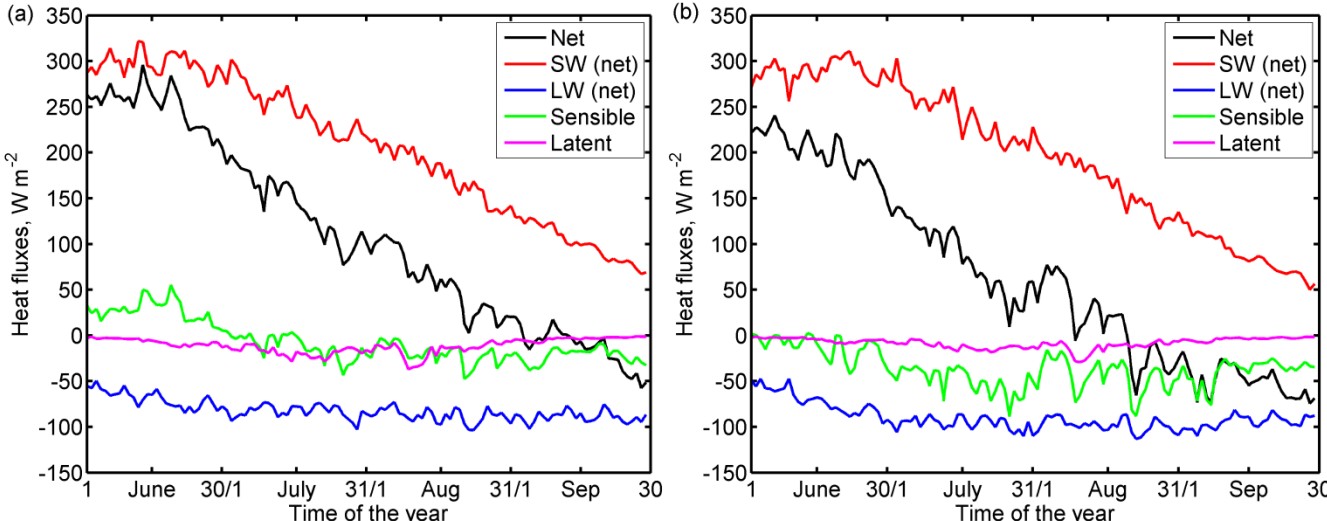

**Figure 7:** The heat balance of a water surface under the influence of atmosphere, W m$^{-2}$: (a) at GS Kusur; (b) at GS Habarova.

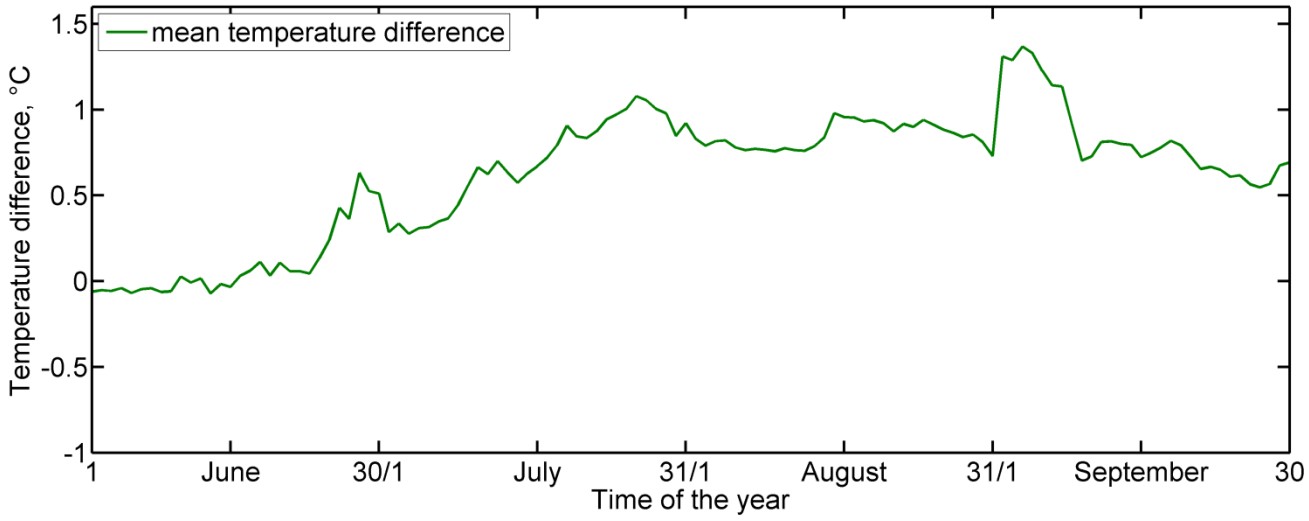

**Figure 8:** The mean difference (2002–2011) between the midstream and near right bank water temperatures at Kusur GS, °C.





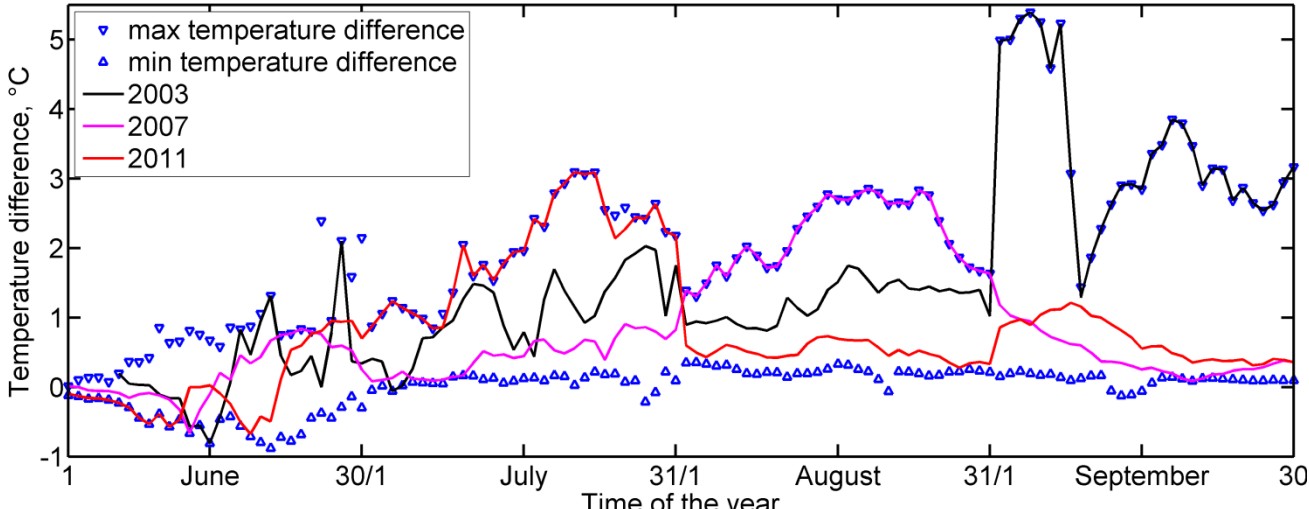

**Figure 9:** The difference between the midstream and near right bank water temperatures at Kusur GS for particular years and maximums and minimums of the temperature differences for the whole period of time from 2002 till 2011, °C.







**Figure 10:** Modeled and observed discharge characteristics for 2012. Top panel: optimized total discharge rate of all tributaries close upstream GS Kusur and mean monthly discharge of Eremeyka River multiplied by 400, m$^3$ s$^{-1}$, pink curve shows the elevation measured at Eremeyka GS, cm. Bottom panel: (a) water temperatures measured and observed at Habarova GS, °C; (b) water temperatures measured and observed at Kusur GS and water temperature observed at Eremeyka GS, °C.





**Figure 11:** Modeled and observed discharge characteristics for 2007. Top panel: optimized total discharge rate of all tributaries close upstream GS Kusur and mean monthly discharge of Eremeyka River multiplied by 400, $m^3\ s^{-1}$. Bottom panel: water temperatures measured and observed at Habarova GS, observed at Kusur GS and at Eremeyka GS, °C, red ellipse indicates the zone where it is impossible to reproduce the temperatures observed at Habarova GS by solving optimization task.



**Table 2**: The mean surface water temperature measured at different gauging stations for June – September from 1981 to 1990.

| Station | Month | | | |
|---|---|---|---|---|
| | **06** | **07** | **08** | **09** |
| **Eremeyka** | 4.41 | 8.41 | 6.39 | 2.2 |
| **Kusur** | 5.49 | 14 | 12.25 | 6.11 |
| **Tit-Ary** | 5.27 | 13.19 | 11.63 | 5.36 |
| **Habarova** | 6.48 | 14.56 | 13.24 | 7.62 |



**Table 3:** The date of the first ice appearance in the fall.

| Station | Year | | | | | | | | | | | | | |
|---|---|---|---|---|---|---|---|---|---|---|---|---|---|---|
| | **1986** | **1987** | **1988** | **1989** | **1990** | **1999** | **2000** | **2001** | **2002** | **2003** | **2004** | **2005** | **2006** | **2007** |
| **Kusur** | 6.10 | 5.10 | 11.10 | 7.10 | 9.10 | 2.10 | 6.10 | 5.10 | 7.10 | 30.09 | 9.10 | 6.10 | 7.10 | 7.10 |
| **Habarova** | 10.10 | 8.10 | 18.10 | 9.10 | 12.10 | 5.10 | 8.10 | 13.10 | 11.10 | 9.10 | 13.10 | 8.10 | 13.10 | 14.10 |





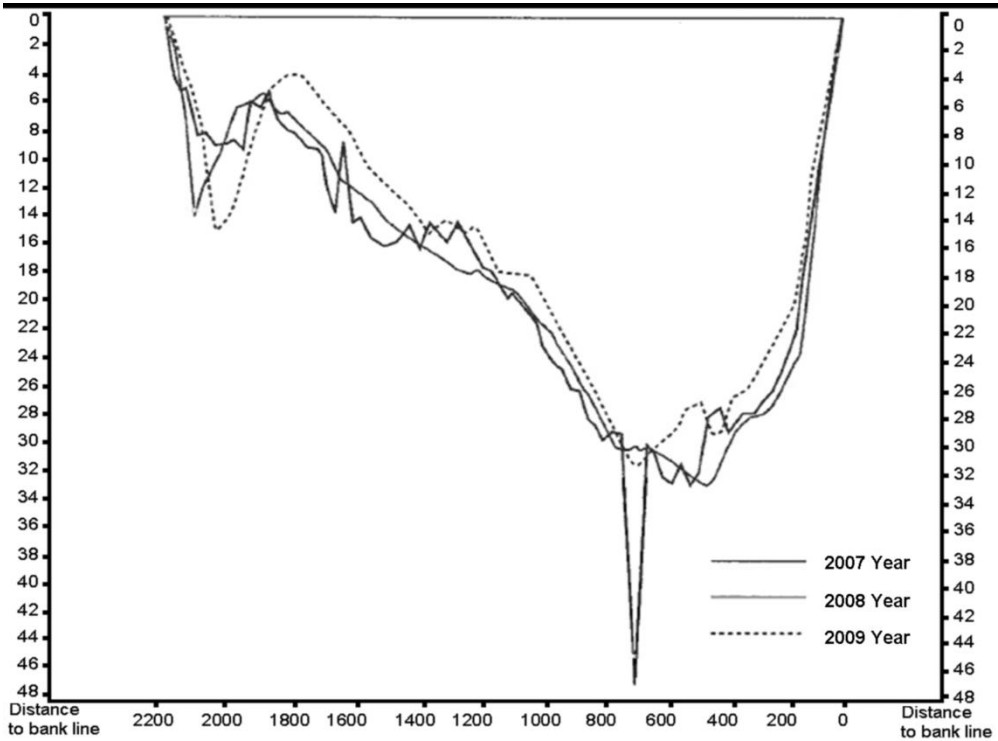

**Figure 12:** The Lena riverbed profile in the area of Habarova GS measured in August different years, main channel, m. The picture is taken from Bolshiyanov et al., 2013.