# Peer review of "The water temperature characteristics of the Lena River at basin"

_Hydrology and Earth System Sciences, 2016_

## Referee Comment (RC1) · Anonymous Referee #1 · 24 Jun 2016

**General comments**

The present paper concentrates on the study of stream temperature records collected at two sites in the lower part of the Lena River basin over the last 60 years. It mostly aims at explaining why temperature measured at the site which is farther north is warmer in summer than temperature measured at the other site, although the opposite would have been expected based on the fact that air temperature is colder at the first site. The authors succeed in identifying the probable causes explaining this phenomenon. Their work is certainly relevant for HESS, although the manuscript requires major revisions in my view. Below are some general points that should be considered by the authors when revising the manuscript.

1. The article should focus more on its main message, i.e. the explanation of the positive temperature difference between Habarova and Kusur. In my opinion, the reason why the temperature difference is considered to be surprising should be explained more clearly and earlier in the manuscript. In addition, the analysis of the temperature trends at Kusur and Habarova over different time periods (Table 2 and lines 13–27 on page 5) divert the reader from the main findings of the paper. These trends are not used in the remaining of the discussion and also do not represent substantial new findings in comparison with the work of Liu et al. (2014). In order to clarify the manuscript, Sections 3 and 4 could be merged into one and subdivided into three subsections: (1) presentation of the data (Fig. 5), (2) presentation of the temperature "anomaly" (Figs. 6 and 7), and (3) discussion of the possible causes for the "anomaly".

2. Regarding the numerical simulations, the use of a regular mesh grid with cells of size $1\,\mathrm{m}\times24\,\mathrm{m}\times100\,\mathrm{m}$ appears to me as highly questionable. From my own experience, the mesh resolution is typically very fine at the walls (i.e. at the banks and over the river bed) and progressively coarsened towards the center of the simulation domain. According to the usual recommendations when using wall functions, the height $z$ of the cells in contact with the walls should be chosen such that the non-dimensionalized height $z^+$ lies approximately between 30 and 500 (Versteeg and Malalasekera, 2007). In the present case, this condition results in $z$ having to be chosen between about $1\,\mathrm{mm}$ and $1\,\mathrm{cm}$, i.e. much smaller than the value of $1\,\mathrm{m}$ used in the manuscript. Although the mesh used by the authors might be possibly sufficient to prove that a cold current forms along the right river bank, I strongly recommend to investigate the impact of the mesh resolution on the numerical solution (see e.g. Shen and Diplas, 2008). In order to spare computational resources, the domain length could be reduced from $20\,\mathrm{km}$ to a bit more than $5\,\mathrm{km}$, since the distance between the tributary located further upstream and Kusur station is about $5\,\mathrm{km}$ (according to Fig. 1).
3. More generally, additional information should be provided about the numerical simulations. In the first numerical experiment, it is not clear whether the Lena River is approximated as a channel with a rectangular cross-section or whether the real bed profile is used. Similarly, nothing is said about the way the lateral tributaries are modeled. Regarding the second numerical experiment, the simulation configuration is missing: which mesh grid is used? Is the river approximated as a rectangular channel? Is Tit-Ary island modeled or not? Figures showing the simulation domain with the numerical results would be a valuable addition to the manuscript.

4. Formulation of the sentences should be revised. Some words are used inappropriately, which makes the manuscript hard to understand. For example, "elevation" is used instead of "water depth" on the first line of page 9, which confused me for a long time. The authors also speak about a "frequency of optimization process" in line 16 of p. 12, which is highly cryptic. Other examples can be found in the Specific Comments below. More generally, it is sometimes hard to understand the point that the authors want to make: statements and counter-statements follow each other, without the authors saying what they think is right (see e.g. point 15 in Specific Comments below). Some facts are stated as if always true, before being proven wrong later in the text. This makes the explanations more difficult to follow and sometimes confuses the reader. Three examples follow:

   - p. 4, ll. 14–27: Based on the observation of vertical stream temperature profiles, the authors state that the "vertical temperature distribution is uniform within [the] cross-section at both Habarova and Kusur". Later, in lines 21–34 on page 7, the authors discuss measurements which prove the exact opposite, namely that "surface water temperature measurements at Kusur GS lack representativeness".
   - p. 13, ll. 5–28: The authors state that the regions of the river bed which

are expected to contribute most to the stream warming are those located in shallow, tranquil flow areas. At the end of the paragraph, they actually prove that the shallow flow area around Island Tit-Ary is colder than expected due to cold water released from the island.

- p. 7, ll. 4–11: According to the authors, "it is obvious that during July–August the heat should be transferred from [the stream] water to the sediments", whereas on p. 14 at ll. 29–31, they conclude that a substantial heat transfer must take place from the sediments to the stream.

**Specific comments**

The following list of specific comments is admittedly long, but should be considered as a set of recommendations as to where the manuscript could be improved.

1. p. 2, ll. 8–9: "the existing analyses of stream temperature [...] are fragmentary and cannot provide the aggregate picture". This sentence is probably overstated and should be moderated.

2. p. 2, l. 13: "These data are rarely used". It is not clear which data have already been used by other authors, and which ones are presented here for the first time. This should be clarified in Sect. 2, where the data is described into more detail.

3. p. 2, ll. 14–17: I recommend to give more explanations on the positive temperature difference observed between Habarova and Kusur, namely briefly explain why this difference is surprising. I would avoid mentioning the station names since they have not been introduced yet: it would be sufficient to indicate that one station is located 200 km north from the other. There is no need to go into too much detail though, since the complete explanation should come later in the text (see General Comments above).

[Figure]

4. p. 2, l. 22: "the water temperatures measured at Kusur [...] reflect the thermal conditions of the Lena River in general". This statement is certainly overstated, as temperature evolves continuously along the river (see Fig. 6). The measurements at Kusur would be at most representative for the lower portion of the Lena basin, but even this fact is actually invalidated in the remaining of the article (see e.g. p. 5, ll. 26–27: "the measurements at Kusur GS [should] be taken for analysis of water temperature changes in the delta head area with a great caution").

5. p. 3, ll. 10–16: A reference to Table 1 is missing, so as to make clear that the temperature data which is described is the same one as in the table.

6. p. 4, l. 6: "fairway" (sic) should be replaced with "river bank" (I assume).

7. p. 4, l. 10: "inflow" should be replaced with "tributary".

8. p. 4, ll. 14–27: This paragraph should be moved later in the text, when the possible causes for the temperature "anomaly" are discussed. As mentioned in the General Comments above, it should also be reformulated so as to be less categorical on the conclusions drawn. On line 25, the Reynolds number should be discussed into more detail: is this number high? What does it imply? Some citations would help readers who are unfamiliar with the Reynolds number.

9. p. 5, ll. 7–27: As mentioned in the General Comments, the authors might consider removing Table 2 and its corresponding paragraph in Sect. 3 so as to focus on the main message of the manuscript.

10. p. 6, ll. 2–3: "The correlation coefficient between maximum events at Habarova GS and Kusur is $\sim 0.6$". It is not clear which quantities were correlated here: was it the magnitude of the events or their time of occurrence? It is also not clear what this correlation demonstrates.

11. p. 6, ll. 5–14: In my opinion, this paragraph is not very clear and should be reformulated. In particular, it should be stated clearly what is meant by the "temperature inconsistency".

12. p. 6, ll. 22–33: Based on Fig. 7, I would actually identify not only one, but three "inconsistencies":

   - In June, the net heat flux is decreasing over time at both Kusur and Habarova, which should imply a decrease in observed stream temperature as well. However, temperature is observed to increase in time at both locations. This might be possibly explained by the upstream conditions, or by unaccounted heat sources.
   - In July and August, the net heat flux is decreasing over the entire reach between Kusur and Habarova, which would imply that the difference in temperature between Habarova and Kusur should decrease as well. Again, the opposite is observed: this "inconsistency" is the one discussed by the authors.
   - In September, the net heat flux is *negative* at both locations, which implies that the temperature at Habarova should be lower than the one at Kusur. Observations contradict this reasoning, which hints at the presence of unaccounted heat sources or measurement errors/non-representativeness (same reasons as those discussed by the authors).

13. p. 7, l. 3: The authors might want to clearly state that ice could be present in June.

14. p. 7, ll. 4–5: "it is obvious that during July – August the heat should be transferred from water to the sediments". This is not obvious to me: either more explanations should be provided, or the sentence should be reformulated.

15. p. 7, ll. 4–17: In both paragraphs d) and e), it is hard to understand the point that the authors want to make. They mention arguments in favor and against each

possible explanation of the temperature difference, without clearly stating in the end whether the explanation could be valid or not. I recommend the authors to reformulate both paragraphs.

16. p. 7, l. 31: "there are no other inflows [. . . ] which could affect the temperature measurements at the stations". The authors could mention the possible presence subsurface water inflows, which cannot be ruled out and are later on shown to be most probably present.

17. p. 8, l. 2: The authors might want to explain in a few words how they obtained the estimate of $0.2\,\mathrm{km}^3$ for the mean annual cumulated discharge of the other tributaries.

18. p. 8, l. 13: The authors use a roughness height of $3.2\,\mathrm{m}$ for the stream bed, but they do not justify their choice. For comparison, Shen and Diplas (2008) and Constantinescu et al. (2014) use a roughness height of $0.01\,\mathrm{m}$, which appears as much more physical to me.

19. p. 8, l. 17 and p. 11, l. 13: Instead of the numbered points, I would make a separate subsection for each numerical experiment. This would clarify the text structure.

20. p. 9, ll. 1–3: The point that the authors want to make is not really clear according to me.

21. p. 9, l. 9: A separate subsection could be created for the analytical computation of mean stream temperature (between the two subsections presenting the numerical experiments, see point 19 above).

22. p. 9, ll. 11: $T_k$ in Eq. (3) is not defined. The authors might consider adding a schema to explain the different terms of the equation.

23. p. 10, ll. 1–20: Figures 8 and 9 could be merged into a single one with two panels (a and b).

24. p. 11, l. 5: It is not clear to me what the high correlation between the water temperature measured at Habarova and the one measured at Eremeyka is supposed to show.

25. p. 11, l. 13: The authors might want to clearly state the goal of the second numerical experiment.

26. p. 11, ll. 15–16: It is not clear to me how the penetration depth of short-wave radiation is used in the model. I encourage the authors to add detailed information regarding the simulation setup (see point 3 in the General Comments above).

27. p. 11, l. 25: It is not clear which "previous estimates of the total discharge" are referred to.

28. p. 12, ll. 1–3: The temperature differences discussed by the authors are difficult to see on the figure since the compared curves are not displayed on a same plot.

29. p. 12, ll. 4–6: The authors invoke the non-representativeness of the measured temperature at Kusur to explain the difference between the measured and modeled temperature curves. They however do not mention the fact that modeling errors could also be responsible for the discrepancy between the two curves. If the simulation setup is similar to the one of the first experiment (i.e. a three-dimensional flow and heat simulation), I would expect the model to be able to provide not only the mean stream temperature at Kusur, but also its actual value at the gauging site location.

30. p. 12, l. 16: The meaning of "larger frequency of optimization process" is not clear to me.

31. p. 12, l. 18: Should not "warming effect" be replaced with "cooling effect"?

[Figure]

32. p. 12, ll. 19–20: The atmospheric forcing do actually not "cool the water from Kusur GS to Habarova in the middle of September", since stream temperature at Habarova is higher than the one at Kusur. On the other hand, the difference between the two temperatures tends to decrease.

33. p. 12, l. 21: "There is an indication in favor of an unaccounted source of heat in the middle of September 2007". The unaccounted heat source is certainly also present in all other years, except that it is more evident in year 2007.

34. p. 13, ll. 32–33: In my opinion, it is not necessary to invoke a "negative heat flux [. . . ] in the area from Kusur to Tit-Ary" to explain the cold temperatures measured at Tit-Ary, since the cold water released by the island is certainly already sufficient enough an explanation.

35. p. 13, l. 32: The authors invoke a "large positive heat flux from the hyporheic zone in the delta head area" to explain the observed temperatures at Habarova. Could not also diffuse subsurface inflows of water be responsible for a warming of the Lena River?

36. p. 20: The lines indicating the respective locations of the gauging sites could be thicker in order to be more visible. The North arrow and the length scale are missing from the figure. I recommend adding an inset showing the entire portion of the Lena River between Kusur and Habarova, in which the locations of the three considered areas (Kusur, Tit-Ary Island and Habarova) would be indicated.

37. p. 21: In the column containing the date of first ice appearance in fall, it is not clear why the measurement period is split in two. For example, could not "1986–1999, 2000–2007" be simply replaced with "1986–2007"? Also, the last row of the table is not discussed in the text and should be removed.

38. p. 22: In my opinion, Fig. 2 is not really necessary and could be removed from the manuscript.

39. p. 24: As mentioned in point 1 in the General Comments, Table 2 could be re-moved.

40. p. 25: In Fig. 5, the mean summer stream temperatures at Kusur in 1936 and 1940 seem to be anomalously high and low, respectively. I encourage the authors to double check the data in these years. Also, the legend of panel (b) is not clear: it took me a long time to figure out that the panel actually displays the observed probability distribution of the time of the year at which the maximum stream temperature is observed.

41. p. 26: The zero line could be indicated in Fig. 7 to improve readability. The authors might also want to remind in the legend that the data correspond to daily mean values, averaged on each summer day between 2002 and 2011.

42. pp. 26–27: As mentioned above, Fig. 8 and Fig. 9 could be regrouped in a single figure. It should also be stated in the legend that the displayed data was computed (and not measured).

43. p. 28: The top panel should be labeled as (a), and the bottom ones as (b) and (c).

44. p. 29: The top panel should be labeled as (a) and the bottom one as (b).

45. p. 30: I would indicate the months using their names instead of numbers 6 to 9.

46. p. 32: The label is missing along the y-axis, and the data units are missing along both axes.

**Technical corrections**

1. p. 1, l. 28: "Costard et al. **(**2007**)**"

2. p. 6, l. 8: "toward**s**"

3. p. 7, l. 4 and l. 12: points d) and e) have been inverted (e is before d)

4. p. 8, l. 5: "2002 to 20**1**1"

5. p. 9, l. 17: Symbol '$>$' should be replaced with '$<$'.

6. p. 13, l. 30: "is" should be removed between "it" and "becomes".

7. p. 13, l. 14, l. 19 and l. 23: "mln" should be replaced with "$\times 10^6$".

8. p. 13, l. 29: "Boike et al. $\underline{(}$2015$\underline{)}$"

9. p. 28, l. 5 and l. 6: "observed" should be replaced with "modeled".

**References**

Shen, Y., and Diplas, P.: Application of two- and three-dimensional computational fluid dynamics models to complex ecological stream flows, Journal of Hydrology, 348(1–2), 195–214, doi: 10.1016/j.jhydrol.2007.09.060, 2008.

Constantinescu, G., Miyawaki, S., Rhoads, B., and Sukhodolov, A.: Numerical evaluation of the effects of planform geometry and inflow conditions on flow, turbulence structure, and bed shear velocity at a stream confluence with a concordant bed, Journal of Geophysical Research: Earth Surface, 119(10), 2079–2097, doi: 10.1002/2014JF003244, 2014.

Liu, B., Yang, D., Ye, B., and Berezovskaya, S.: Long-term open-water season stream temperature variations and changes over Lena River Basin in Siberia, Global and Planetary Change, 48(1–3), 96–111, doi: 10.1016/j.gloplacha.2004.12.007, 2005.

Versteeg, H., and Malalasekera, W.: An Introduction to Computational Fluid Dynamics: The Finite Volume Method, Prentice Hall, 2007.

---

## Referee Comment (RC2) · Anonymous Referee #2 · 1 Jul 2016

**Review of :The water temperature characteristics of the Lena River at basin outlet in the summer period. Paper** # hess-2016-254

Authors: Vera Fofonova, Igor Zhilyaev, Marina Kraineva, Dina Iakshina, Nikita Tananaev, Nina Volkova, and Karen H. Wiltshire

General Comments

The purpose of the paper appears to be two-fold. One, is to explain the perceived anomalous increases in observed water temperature from the site at Kusur to the site at Habarova (Khabarova) (Page 2, Lines 14-17). The second, it seems, is to determine whether or not the long-term observations at Kusur represent the mean stream temperature of the Lena River (Page 4 Line 15). Given the size of the Lena River's watershed and the associated estimated average annual streamflow ($\sim$17,000 m$^3$/sec), knowledge of the river's thermal energy budget would seem to be important, most notably if the Earth's temperature increases, as most climate models predict. This is certainly consistent with the goals of the journal.

However, while acknowledging the need for accurate estimates of both flow and water temperature, there are significant challenges to doing so in this case given the need to synthesize, or, in the words of the authors, "optimize", flow data, the crude manner in which the earliest field observations of water temperature were made and the poor choice of a stream temperature measurement location at Kusur. Application of a numerical model to enhance available observations is consistent with the notions of Bayesian analysis and state estimation. In this case, however, the task is daunting. The authors are left to speculating on many processes that affect the thermal energy budget including streamflow dynamics, river–atmosphere heat exchange and streambed heat transfer. This leads to a litany of apologies by the authors for the high degree uncertainty in their analysis. Matters are made worse in the paper due to poor grammar, questionable logic, and missing information. I have given some examples below of the structural and scientific issues associated with this paper.

Given the faulty design and lack of quality assurance of the monitoring program, a qualitative analysis of the data is as conclusive as that given in paper. The observation record at Habarova, though not quite as long as that at Kusur, is still quite lengthy. It is closer to the mouth of the Lena River and, hence, more representative of the transport of thermal energy to the Laptev Sea. In addition, it does not appear to be influenced by the proximity of input from tributaries, although this is not obvious from the paper. What is needed here at this stage,

rather than the application of a complex mathematical model using questionable inputs, is the development of an appropriate experimental design. Anonymous reviewer #1's thorough analysis details the many technical difficulties in this paper. It is difficult to see how they might be corrected without major revisions.

Specific Comments

| Page | Line #'s | |
|---|---|---|
| 2 | 15 | Use of acronyms like "GS" for common nouns like "gauging station" is not standard. |
| 3 | 5 | "web source" is not a recognized reference. There are numerous occurrences in the paper. |
| 3 | 10 | The description of the monitoring frequency is unclear. |
| 3 | 15 | An unorthodox measurement technique with no quality assurance. |
| 4 | 6 | A "fairway" in the US is on a golf course. What information does "The left bank is shallow" add? |
| 5 | 9 | Do authors mean "presence of a trend" rather than "presence of trend"? Numerous occurrences of the missing article, "a". |
| 5 | 14 | Do authors mean "consider the period" rather than "consider period"? Numerous occurrences of the missing article, "the". |
| 5 | 21 | "Sic" ? |
| 5 | 30 | "for example, are close" rather than "for example, close"? |
| 6 | 4 | "bootstrap analysis" is not explained. |
| 6 | 21 | "water temperature still can increasing"? |
| 7 | 12-13 | "The possible reason for this puzzling disagreement could be the non-representativeness of measurements at one or both the stations". Agreed. This is an |
| 8 | 10 | The description of the model, "COMSOL", is inadequate. |
| 8 | 12 | The description of the term, "wall function", is inadequate. |
| 8 | 30-35 | Confusing. |

| 9  | 8-9   | "It is highly expected due to the use of wall functions."? |
|----|-------|-----------------------------------------------------------|
| 9  | 10-15 | The description of Equations (1)-(3) is inadequate. |
| 10 | 33    | "Proving" is not the correct verb here. |
| 11 | 21    | What does "Optimization" mean here, and how was it done? |
| 12 | 28    | Define "talik". |

These comments are by no means exhaustive. Rather they give a flavor of the many editorial and scientific issues associated with this paper.

---

## Editor Comment (EC1) · B. Schaefli (Editor) · 4 Jul 2016

This manuscript has been reviewed in detail by two reviewers. Both came to the conclusion that the scientific relevance of the paper is good but that the scientific quality and the presentation quality are fair to poor. One reviewer recommends rejection, one major revisions.

In light of the above and of the detailed reviewer comments, I cannot recommend the revision of this paper for a publication in HESS. The authors should nevertheless answer the reviews in the public discussion, which will in particular be the basis for my recommendation to resubmit a substantially revised version.

---

## Author Comment (AC1) · 18 Jul 2016

Dear Reviewer, We are very grateful for the valuable and helpful comments. Please, find in Supplement the answers and updated version of the manuscript. The changes in the manuscript have been marked in blue. Yours sincerely, Vera Fofonova, on behalf of the authors

Please also note the supplement to this comment:
http://www.hydrol-earth-syst-sci-discuss.net/hess-2016-254/hess-2016-254-AC1-supplement.zip

---

## Author Comment (AC2) · 18 Jul 2016

Dear Reviewer,

Please, find in the Supplement our answers to the comments and updated version of the manuscript. The changes in the manuscript have been marked in blue.

Yours sincerely,

Vera Fofonova, on behalf of the authors

Please also note the supplement to this comment:
http://www.hydrol-earth-syst-sci-discuss.net/hess-2016-254/hess-2016-254-AC2-supplement.zip

---

## Author Comment (AC3) · 18 Jul 2016

Dear Editor,

We have changed the manuscript according to the comments given by Reviewers. Especially, we would like to thank the first Reviewer for helpful suggestions and valuable comments.

Yours sincerely,

Vera Fofonova, on behalf of the authors

---

## Referee Comment (RC3) · Anonymous Referee #2 · 3 Aug 2016

**Review of: The water temperature characteristics of the Lena River at basin outlet in the summer period. Paper # hess-2016-254**

Response to Authors: Vera Fofonova, Igor Zhilyaev, Marina Kraineva, Dina Iakshina, Nikita Tananaev, Nina Volkova, and Karen H. Wiltshire

The basic problems with this paper remain. It still contains the kind of grammatical errors that were rife in the original. It does not sufficiently describe or explain critical issues:

• 'wall function' – While the term is familiar to many, T.J. Craft of the Manchester School of Mechanical Aerospace and Civil Engineering mentions fourteen different approaches in an article on Wikipedia. For purposes of doing an adequate review, as well as for the general readership of HESS, a specific reference (there are three in the paper) and an equation are necessary. This particularly important since it appears that the wall function the authors chose, provides the basis for the simple model embodied in Eq. 1-3.

• 'bootstraping' (generally spelled with two p's, as in 'bootstrapping') –A generic statistical technique, as described in Wikipedia article: "In statistics, **bootstrapping** can refer to any test or metric that relies on random sampling with replacement". The reference provided in the paper links to a lecture comparing the results from using aspirin compared to using a placebo. If this is an important element in the paper (it's not clear that it is) an explanation is in order.

• 'optimization' – There is a gesture in the direction of an explanation of this in the revised manuscript, but it's one that really doesn't provide enough detail for the reviewer or general readership. Furthermore, its application is highly questionable as a method for forcing the model to replicate the observations. In this regard, it is hard to believe that the high frequencies in the hydrographs resulting from the 'optimization', as shown in Figures10 and 11, bear a relationship to reality in the Lena River system. When solving an inverse geophysical problem in a highly underdetermined problem, as this is, an exercise of this kind reveals little about the actual behavior of the system.

• 'quality assurance and experimental design' – Making sense of the long record at the Kusur gaging station is a worthy goal. The challenges associated with the location of the temperature monitoring site and the measurement method make it important that there be a well-designed experimental design. For purposes of evaluating the previous record, one made with questionable measurement methods, some effort should be made to characterize the uncertainty of the observations.

Finally, after 16 pages, the authors can only conclude, (1) "The difference in the behaviour of stream temperatures at Habarova GS (*sic*) and Kusur and non-representativeness of the measurements at Kusur GS (*sic*) for the whole cross-section indicate that the measurements at Kusur GS (*sic*) should be taken for analysis of water temperature changes in the delta head area with a great caution", and, (2) "There are indications in favour of an unaccounted source of heat in the late summer/beginning of fall from the riverbed to the water in the area of the delta head. More analysis and observations are required to make further statements in this direction".

It is difficult to see how this paper in its present form significantly increases our knowledge of the dynamics of stream temperature in the Lena River system.

---

## Author Comment (AC4) · 5 Aug 2016

Dear Reviewer,

Please, find the answers and updated manuscript in the attached folder.

Yours sincerely,

Vera Fofonova, on behalf of the authors.

Please also note the supplement to this comment:
http://www.hydrol-earth-syst-sci-discuss.net/hess-2016-254/hess-2016-254-AC4-supplement.zip

---

## Author Comment (AC5) · 5 Aug 2016

Dear Editor,

Please, take into account that we do not agree with the evaluation of our work by Reviewer 2. We have done a really extensive work on the problem and obtained important results from our point of view, which can be interesting for many people:

-one of the main reason of the anomalously high water temperature at Habarova Station is found and explained using unique database

-quantitative and qualitative assessments of the impact of small rivers upstream Kusur on the measurements at Kusur Station are given

-trend estimates and behaviors of the water temperature at two basin outlet stations

are given and discussed

-important gaps for the system understanding are indentified

Unfortunately, the comments of Reviewer 2 do not provide clear points what should be improved in the manuscript and signalize about lack of understanding of the manuscript (please, see the answers). We understand that our manuscript has open end and that it is not a final dot in the problem understanding. There are several issues that we cannot resolve using existing database and numerical instruments without additional measurements. However, the manuscript gives base for the future measurements and sets its directions.

Yours sincerely,

Vera Fofonova, on behalf of the authors